# Improved Thermal Anisotropy of Multi-Layer Tungsten Telluride on Silicon Substrate

**DOI:** 10.3390/nano13121817

**Published:** 2023-06-07

**Authors:** Mengke Fang, Xiao Liu, Jinxin Liu, Yangbo Chen, Yue Su, Yuehua Wei, Yuquan Zhou, Gang Peng, Weiwei Cai, Chuyun Deng, Xue-Ao Zhang

**Affiliations:** 1College of Physical Science and Technology, Xiamen University, Xiamen 361005, China; 19820201153527@stu.xmu.edu.cn (M.F.); xiaoliu@stu.xmu.edu.cn (X.L.); jxliu@stu.xmu.edu.cn (J.L.); bobbychen@stu.xmu.edu.cn (Y.C.); 19820190154690@stu.xmu.edu.cn (Y.S.); yqzhou@stu.xmu.edu.cn (Y.Z.); 2College of Science, National University of Defense Technology, Changsha 410073, China; penggang@nudt.edu.cn; 3College of Advanced Interdisciplinary Studies, National University of Defense Technology, Changsha 410073, China; weiyuehua18@nudt.com; 4Jiujiang Research Institute of Xiamen University, Jiujiang 332105, China

**Keywords:** substrate coupling, thermal anisotropy, laser-heating Raman thermometry, electrical-heating Raman thermometry

## Abstract

WTe_2_, a low-symmetry transition metal dichalcogenide, has broad prospects in functional device applications due to its excellent physical properties. When WTe_2_ flake is integrated into practical device structures, its anisotropic thermal transport could be affected greatly by the substrate, which matters a lot to the energy efficiency and functional performance of the device. To investigate the effect of SiO_2_/Si substrate, we carried out a comparative Raman thermometry study on a 50 nm-thick supported WTe_2_ flake (with κ_zigzag_ = 62.17 W·m^−1^·K^−1^ and κ_armchair_ = 32.93 W·m^−1^·K^−1^), and a suspended WTe_2_ flake of similar thickness (with κ_zigzag_ = 4.45 W·m^−1^·K^−1^, κ_armchair_ = 4.10 W·m^−1^·K^−1^). The results show that the thermal anisotropy ratio of supported WTe_2_ flake (κ_zigzag_/κ_armchair_ ≈ 1.89) is about 1.7 times that of suspended WTe_2_ flake (κ_zigzag_/κ_armchair_ ≈ 1.09). Based on the low symmetry nature of the WTe_2_ structure, it is speculated that the factors contributing to thermal conductivity (mechanical properties and anisotropic low-frequency phonons) may have affected the thermal conductivity of WTe_2_ flake in an uneven manner when supported on a substrate. Our findings could contribute to the 2D anisotropy physics and thermal transport study of functional devices based on WTe_2_ and other low-symmetry materials, which helps solve the heat dissipation problem and optimize thermal/thermoelectric performance for practical electronic devices.

## 1. Introduction

Low-symmetry materials exhibit anisotropic thermal [1,2,3,4], electrical [5,6,7,8], optical [5], optoelectronic [9], and thermoelectric [10] properties along different lattice orientations, which provides a brand-new opportunity to design high-performance devices. WTe_2_, a recently popular player in two-dimensional (2D) materials with low-symmetry lattice structures, has exhibited outstanding functional device applications [11,12] based on its heavy atomic mass, low-energy optical absorption [13], thickness-dependent anisotropy of Raman modes [14], and superconductivity properties [15,16]. It has been reported that the conductivity and the photoelectric anisotropy ratio of WTe_2_ film are about 10^3^ and 300 [17], respectively, allowing it to be applied to anisotropic electric and photonic devices [18,19,20,21,22].

Considering that heat dissipation efficiency is crucial to device reliability and performance, while the thermal transport characteristics of WTe_2_ and other low-symmetry materials are quite sensitive to varying external conditions, it is necessary to explore the effect of external conditions before designing WTe_2_-based functional devices. The ambient temperature, mechanical strain, and substrate coupling will all lead to diverse thermal transport behaviors, which may affect thermal anisotropy and device performance [1,23,24]. In practical applications, especially low-symmetry materials are mostly integrated on substrates; therefore, understanding the effect of substrate is important for device design. For example, the thermal conductivity of suspended black phosphorene (BP) flake exhibits obvious anisotropy along different directions [1,2,10,25,26]. When supported on a solid substrate, the thermal anisotropy of BP flake will show obvious improvement compared to the suspended one, which is proved by molecular dynamics simulations [27].

As most 2D-integrated circuits are silicon-based, we chose an SiO_2_/Si substrate to explore the role of substrate coupling on multilayer WTe_2_. In this paper, the in-plane thermal conductivity of 50 nm-thick suspended and supported WTe_2_ samples along zigzag/armchair axes is investigated by laser-heating and electrical-heating Raman thermometry. The experimental results show that the in-plane thermal conductivity of supported WTe_2_ flake is κ_zigzag_ = 62.17 W·m^−1^·K^−1^, κ_armchair_ = 32.93 W·m^−1^·K^−1^, and that of suspended WTe_2_ flake is κ_zigzag_ = 4.45 W·m^−1^·K^−1^, κ_armchair_ = 4.10 W·m^−1^·K^−1^. Low-frequency phonons mainly contribute to the thermal conductivity along zigzag and armchair directions [28], and the thermal conductivity along the zigzag direction is greater for WTe_2_ flakes. In addition, it has been reported that the mechanical properties of materials will be affected when mechanical force is applied [29]. It is possible that the different contributions of low-frequency phonons and mechanical properties along zigzag and armchair directions caused by substrate have led to a significantly larger thermal anisotropy ratio for supported WTe_2_ flake (κ_zigzag/_κ_armchair_ ≈ 1.89) compared to the case for suspended WTe_2_ flake (κ_zigzag/_κ_armchair_ ≈ 1.09). Therefore, it is suggested that low-frequency phonons and mechanical properties are two of the possible reasons for the variation in the thermal anisotropy ratio of supported WTe_2_ flake. Our finding may be helpful to study the heat dissipation process in low-symmetry materials and offer guidance for efficient thermal management of low-symmetry material devices.

## 2. Materials and Methods

### 2.1. Preparation of WTe_2_ Samples

We first placed a small piece of polydimethylsiloxane (PDMS) on a clean slide and set it aside. A piece of bulk WTe_2_ (PDMS and bulk WTe_2_ were purchased from Shanghai Onway Technology Co., Ltd., Shanghai, China, http://www.onway-tec.com/, accessed on 7 September 2021) was placed on the scotch tape and folded repeatedly. The tape was then attached to the PDMS and pressed gently to make it fit closely to the PDMS. After ten seconds, the tape was quickly removed from the PDMS, and WTe_2_ flakes were transferred from the tape to the surface of the PDMS. We selected a uniform area of WTe_2_ flake on PDMS, then transferred it to the silicon substrate with periodic holes (with a diameter of 12 μm). The WTe_2_ flake covered the silicon holes to construct the suspending region of the WTe_2_ sample.

We took another piece of bulk WTe_2_ on a clean scotch tape and repeatedly folded the tape, then attached it to an SiO_2_ (285 nm)/Si substrate and pressed it gently so that the tape fit tightly to the substrate. After five minutes, the tape was slowly removed from the substrate, and WTe_2_ flakes with different optical contrasts covered the surface of the substrate, which could be observed through an optical microscope. We selected an area of WTe_2_ flake, then designed specific electrode patterns along zigzag and armchair directions of WTe_2_ flake by electron-beam lithography (Raith e-LINE Plus, Dortmund, Germany), and deposited Ti (with a thickness of 5 nm) and Au (with a thickness of 70 nm) on the surface of WTe_2_ flake by electron-beam evaporation (Kurt J. Lesker PVD75, Pittsburgh, PA, USA) to complete the preparation of the WTe_2_ device (supported WTe_2_ sample).

### 2.2. Characterizations of WTe_2_ Samples

Optical microscopy imaging was performed by the LV100D system (Nikon, Japan). The thickness and uniformity of samples were obtained by AFM (NT-MDT Company, Moscow, Russia). During the AFM measurement, there is an interaction force between the probe tip and the surface atoms of the sample. The interaction force is weaker when the distance between the tip and the sample surface is greater. When the tip is closer to the sample surface, the interaction force is greater. This change in interaction force causes a deformation of the cantilever beam, which is detected by the photosensitive detector and fed back to finally obtain the surface information of the sample.

Raman spectroscopy is a common tool to characterize the lattice structure, electrical, optical, and phonon properties of 2D materials [30,31,32]. Angle-dependent Raman spectroscopy was extensively used to study anisotropic lattice structures [33,34,35]. In this paper, we investigate the optical anisotropy of WTe_2_ samples by angle-dependent Raman spectroscopy (Renishaw, Wotton-under-Edge, UK) with a 532 nm excitation laser with 50× objective (N_A_ = 0.55, and the laser spot size is about 616 nm), and the samples were rotated by a sample stage. Appendix A shows the schematic diagram of the polarized-Raman configuration. During the characterization of angle-dependent Raman spectroscopy, we fixed the incident laser polarization (e_i_) parallel to the scattered laser polarization (e_s_) (labeled as e_i_ ‖ e_s_, as shown in Appendix A) and set θ as the angle between the incident laser polarization (e_i_) and the zigzag axis. We placed the zigzag axis of the WTe_2_ sample parallel to the incident laser polarization (defined as 0°) [34,35], and the sample could be rotated clockwise from 0° to 360° by a sample stage. The optical anisotropy of the WTe_2_ sample is then investigated by analyzing the variation of Raman intensity with θ. The details of angle-dependent Raman spectroscopy are presented in the illustrations in Figure 1 and Appendix A.

### 2.3. Raman Thermometry Measurements of WTe_2_ Samples

Raman thermometry is an important method for investigating the thermal conductivity of 2D materials [36,37,38]. As the temperature increases, the Raman peak frequency will shift towards the lower frequency, which means that the Raman peak is redshifted [37]. The thermal conductivity can be obtained from the change in Raman frequency [39]. The thermal conductivity of WTe_2_ samples is investigated by Raman thermometry in the following two steps [28,40,41]. First, the in situ Raman spectra are obtained under a certain temperature range, from which the coefficient between Raman peak frequency and temperature can be extracted. The temperature-dependent in situ Raman spectra are performed with a temperature range from liquid nitrogen to room temperature to avoid damage to the sample [28,40,41]. Second, the sample is heated by increasing the laser power/bias voltage to analyze the Raman peak frequency shift against changing power. Laser power-dependent and bias voltage-dependent in situ Raman spectra are performed with low laser power and bias voltage range at room temperature [28,40]. Here, based on the different heating sources, Raman thermometry can be divided into laser-heating Raman thermometry (heating the sample with laser power) [41] and electrical-heating Raman thermometry (heating the sample with electrical power) [40].

The thermal conductivity of suspended WTe_2_ flake along zigzag and armchair directions is measured by laser-heating Raman thermometry with temperature-dependent (Renishaw, Wotton-under-Edge, UK) and laser-dependent in situ Raman spectra (Alpha 300R system, WITec Company, Ulm, Germany). The temperature is controlled by the A599 heating accessory holder (Bruker Corporation, Billerica, MA, USA). The in-plane thermal conductivity along different directions could be calculated by the equation $\kappa_{\mathrm{zigzag}}=\chi_{T,\;\mathrm{zigzag}}\;(1/2\pi h)\;{(\delta \omega_{\mathrm{zigzag}}/\delta P_{A,\;\mathrm{zigzag}})}^{-1}$ and $\kappa_{\mathrm{armchair}}=\chi_{T,\;\mathrm{armchair}}\;(1/2\pi h)\;{(\delta \omega_{\mathrm{armchair}}/\delta P_{A,\;\mathrm{armchair}})}^{-1}$, where $\chi_{T}$ is the first-order temperature coefficient, h is the thickness, P_A_ is the absorbed laser power of suspended WTe_2_ flake (P_A_ = AP, A is the absorptivity and P is the incident laser power), and δω is the Raman peak position shift caused by the increase in absorbed laser power δP_A_.

The thermal conductivity of supported WTe_2_ flake along zigzag and armchair directions is measured by electrical-heating Raman thermometry with temperature-dependent (Renishaw, Wotton-under-Edge, UK) and bias voltage-dependent in situ Raman spectra (Alpha 300R system, WITec Company, Ulm, Germany). The electrical measurement is measured by Keithley 4200. Based on the relationship between the temperature and the channel power density, we could calculate the in-plane thermal conductivity along different directions of supported WTe_2_ flake by the equation $T_{\mathrm{zigzag}\;}=T_{0,\;\mathrm{zigzag}}+P_{\mathrm{zigzag}}/\kappa_{⊥}$ and $T_{\mathrm{armchair}}={\;T}_{0,\;\mathrm{armchair}}+P_{\mathrm{armchair}}/\kappa_{⊥}$, where T is the temperature of supported WTe_2_ flake, κ_⊥_ is that along the direction perpendicular to current, P is the power density per channel length, and T_0_ is the temperature when the bias voltage is 0 V.

## 3. Results and Discussion

Figure 1a shows the low-symmetry crystal structure of WTe_2_. The tellurium-tungsten-tellurium atomic planes exhibit an orthorhombic lattice through vertical stacking along the *z*-axis by van der Waals force. The tungsten atoms form a W-W chain along the *x*-axis, and the upper tellurium layer is rotated 180° with respect to the bottom tellurium layer, showing a clear anisotropic structure. We define the *x*-axis as a zigzag direction and the *y*-axis as an armchair direction. Since the cleave energy in the zigzag direction is smaller than that in the armchair direction, bulk WTe_2_ will easily break along the zigzag direction during mechanical exfoliation [28] (it is observed that the edge along the zigzag direction is longer through the optical microscope).

We then explore the optical anisotropy of multilayer WTe_2_ by angle-dependent Raman spectroscopy, which contributes to the subsequent investigations of thermal transport properties along different lattice directions. The sample was then rotated clockwise from 0° to 360° during angle-dependent Raman measurements in a parallel-polarized configuration (labeled as $e_{i}∥e_{s})$, where the incident 532 nm laser polarization (e_i_) was parallel to the scattered laser polarization (e_s_). Figure 1b shows the contour map of Raman intensity varying as a function of angle θ (the angle between laser polarization and zigzag direction) for three A_1_ modes located at about 133 cm^−1^ (^4^A_1_), 163 cm^−1^ (^8^A_1_), and 212 cm^−1^ (^10^A_1_).

According to the Placzek approximation [42,43], the variation of Raman intensity with θ for the A_1_ and A_2_ modes mentioned above can be understood. As θ ranges from 0° to 360°, the Raman intensity of A_1_ and A_2_ modes can be described by the following equation:(1)$${I\propto \left|e_{i}\cdot \tilde{R}\cdot e_{s}\right|}^{2}$$

In a parallel-polarized configuration, the unitary vector of incident light is e_i_ = (cos θ, sin θ, 0) and that of scattered light is e_s_ = (cos θ, sin θ, 0). $\tilde{R}$ is the Raman tensor. $\tilde{R}\left(A_{1}\right)=\left(\begin{array}{ccc} a & 0 & 0\\ 0 & b & 0\\ 0 & 0 & c \end{array}\right)$ and $\tilde{R}\left(A_{2}\right)=\left(\begin{array}{ccc} 0 & d & 0\\ d & 0 & 0\\ 0 & 0 & 0 \end{array}\right)$. Then the Raman intensities of A_1_ and A_2_ modes become functions of θ, as is described [33]
(2)$$I_{A1}^{∥}\propto a^{2}{\left|1+\left(\frac{b}{a}-1\right)\sin^{2}\theta \right|}^{2},$$
(3)$$I_{A2}^{∥}\propto d^{2}\sin^{2}2\theta ,$$
where a, b, and d are the elements of Raman tensors that determine the peak intensity. Thus, the Raman intensity of A_1_ mode exhibits two-fold symmetry, while that of A_2_ mode exhibits four-fold symmetry, in accordance with our experiment results as shown in Figure 1c–f. Equation (2) exhibits that the variation of Raman intensity of A_1_ mode with θ is related to the relationship between a and b. For a>b, the maximum Raman intensity value appeared at θ = 0° and θ = 180°, which means the incident light polarization is parallel to W-W chains (zigzag direction), such as in ^4^A_1_ and ^8^A_1_ modes. Conversely, the maximum Raman intensity is located at θ = 90° and θ = 270° (armchair direction), when the incident light polarization is perpendicular to W–W chains for a<b, namely, ^10^A_1_ mode.

After characterizing the anisotropic crystal structure and Raman modes of WTe_2_ flake, we then investigate the in-plane anisotropy of thermal transport. As WTe_2_ flakes are mostly integrated into silicon-based electronic devices, it is reasonable to investigate the thermal transport properties of WTe_2_ flakes supported on SiO_2_/Si substrates for actual applications. Here, we explored the anisotropic thermal transport of both suspended and supported WTe_2_ samples through laser-heating and electrical-heating Raman thermometry.

The height profile in Appendix A clearly indicates that the thickness of suspended WTe_2_ flake is about 50 nm. After basic characterization, temperature- and laser power-dependent Raman spectra were used to study the anisotropic thermal transport of suspended WTe_2_ flake. Figure 2b shows the schematic diagram of the laser-heating Raman thermometry setup, where the laser (focused on the center of suspended WTe_2_ flake) is used to heat suspended WTe_2_ sample.

Appendix A shows the peak positions of prominent Raman modes ^8^A_1_ (located at about 163 cm^−1^) and ^10^A_1_ (located at about 212 cm^−1^) as a function of temperature (ranging from 203 K to 273 K, and the step is 10 K). It was found that ^8^A_1_ and ^10^A_1_ modes exhibit apparent redshifts as the temperature increases due to anharmonic lattice vibrations and thermal expansion [44]. Where the frequencies of ^8^A_1_ Raman mode along the zigzag and armchair directions are redshifted by 0.9025 cm^−1^ and 0.748 cm^−1^, and the frequencies of ^10^A_1_ Raman mode along the zigzag and armchair directions are redshifted by 1.2535 cm^−1^and 0.821 cm^−1^, respectively. As reported in a previous study, ^10^A_1_ mode was more sensitive to temperature than ^8^A_1_ mode [14,28], so we chose ^10^A_1_ mode as the thermometer. Figure 2c indicates that the Raman peak position for ^10^A_1_ mode follows a linear tendency with increasing temperature along zigzag and armchair directions, as fitted by the equation [45]
(4)$$\omega \left(T\right)=\omega_{0}+\chi_{T}T,$$
where $\omega_{0}$ is the Raman peak position at 0 K, $\chi_{T}$ is the first-order temperature coefficient, and T is the temperature. According to Figure 2c, the $\chi_{T}$ of ^10^A_1_ mode can be extracted, $\chi_{T,\;\mathrm{zigzag}}$ = −0.0177 ± 0.0008 cm^−1^·K^−1^ and $\chi_{T,\;\mathrm{armchair}}$ = −0.0126 ± 0.0004 cm^−1^·K^−1^, which is within the same order of magnitude as the data of previous studies [28].

To calculate the thermal conductivity of suspended WTe_2_ flake, laser-dependent Raman spectra were further investigated in detail. The laser spot size (about 616 nm) is much smaller than the suspended area of WTe_2_ flake on silicon substrate (the diameter of the holes on silicon substrate is 12 μm), meaning the heat transfer to the substrate is negligible. The corresponding Raman spectra of suspended WTe_2_ flake as the incident laser power ranges from 106 μW to 325 μW are shown in Appendix A, suggesting the redshifts of ^8^A_1_ and ^10^A_1_ modes. Where the frequencies of ^10^A_1_ Raman mode along the zigzag and armchair directions are redshifted by 1.402 cm^−1^ and 0.8485 cm^−1^. The linear relationship between the Raman peak position for ^10^A_1_ mode and increasing laser power is shown in Figure 2d, as described by the equation [37]
(5)$$∆\omega =\omega \left(P_{2}\right)\;-\;\omega \left(P_{1}\right)=\chi_{P}\left(P_{2}\;-\;P_{1}\right)={\;\chi }_{P}∆P,$$
where $\chi_{P}$ is the first-order laser-dependent coefficient and P is the laser power. For suspended WTe_2_ flake, $\chi_{P,\;\mathrm{zigzag}}$ = −0.0069 ± 0.0002 cm^−1^·μW^−1^ and $\chi_{P,\;\mathrm{armchair}}$ = −0.0053 ± 0.0001 cm^−1^·μW^−1^. The thermal conductivity κ of suspended WTe_2_ flake can be calculated from [36]
(6)$$\frac{\partial Q}{\partial t}=-\kappa \oint \nabla T\cdot \mathrm{dS},$$
where Q is the heat transferred along the cross-section area S (during the time t), and T is the temperature. Considering the radial heat dissipation from the center to the edge of the suspended area, Equation (6) is converted to κ=(1/2πh) (∆P/∆T), where h is the thickness of suspended WTe_2_ flake. ΔT corresponds to the temperature change that is caused by ΔP. By analyzing the linear relationship between laser power and increasing temperature, the thermal conductivity can be calculated from
(7)$$\kappa =\chi_{T}\left(\frac{1}{2\pi h}\right){\left(\frac{\delta \omega }{\delta P_{A}}\right)}^{-1},$$
where P_A_ is the absorbed laser power of suspended WTe_2_ flake (P_A_ = AP, A is the absorptivity and P is the incident laser power). δω is the Raman peak position shift caused by the increase in absorbed laser power δP_A_ [36,37]. Appendix A shows the angle-dependent absorbance plot of suspended WTe_2_ flake at room temperature (the measured wavelength is 532 nm). Considering the influence of CCD noise and dark current on the experimental data, the uncertainty of absorptivity is about 0.10%. As a result, the extracted thermal conductivity along the zigzag direction, κ_zigzag_ = 4.45 ± 0.20 W·m^−1^·K^−1^, is about 1.09 times that along the armchair direction, κ_armchair_ = 4.10 ± 0.13 W·m^−1^·K^−1^, showing an obvious thermal anisotropy. Such anisotropy may be attributed to the phonons having different mean free paths along zigzag and armchair directions [28], which has also been observed in BP and Ta_2_NiS_5_ [3,41].

As a common substrate for practical applications, SiO_2_/Si substrate might impact the thermal transport of the WTe_2_ flake above. Hence, we investigate the thermal properties of multilayer WTe_2_ supported on SiO_2_/Si substrate by electrical-heating Raman thermometry. Appendix A shows the similar thickness of supported and suspended WTe_2_ samples. Figure 3a,b shows the optical image of supported WTe_2_ flake and the schematic diagram of electrical-heating Raman thermometry. We also characterize the electrical transport properties of supported WTe_2_ flake with the bias voltage changing from −3 V to 3 V (Appendix A). First, the orientation-dependent output curves along zigzag and armchair directions reflect good ohmic contact between WTe_2_ flake and metal electrodes. Furthermore, the electrical power density P (power density per length) gradually increases with the voltage ranging from 0 V to 3 V. According to previous literature [40], P = F × I_DS_ = (I_DS_ × V_DS_)/L, where F means the bias voltage per channel length and I_DS_ means source/drain current. Therefore, we can calculate the thermal conductivity of supported WTe_2_ flake by analyzing the in situ Raman spectra under different bias voltages.

Appendix A shows the temperature-dependent and bias voltage-dependent Raman spectra of supported WTe_2_ flake. The curve in Figure 3c indicates that the Raman peak position for ^10^A_1_ mode exhibits a linear redshift with increasing temperature (ranging from 213 K to 293 K, and the step is 10 K). According to Equation (4), for the ^10^A_1_ mode of supported WTe_2_ flake, $\chi_{T,zigzag}$= −0.0153 ± 0.0002 cm^−1^·K^−1^ and $\chi_{T,armchair}$ = −0.0119 ± 0.0004 cm^−1^·K^−1^ as in Figure 3c. Then, we analyze the Raman spectra when electrical bias is applied along different directions of supported WTe_2_ flake. As the bias voltage ranges from 0 V·μm^−1^ to 0.2 V·μm^−1^ (with 0.02 V/μm step), the electrical power density reaches 3 mW·μm^−1^. Figure 3d shows the Raman peak position of ^10^A_1_ mode as a function of electrical power density, where the redshift along the zigzag direction is 0.6225 cm^−1^ and that along the armchair direction is 0.223 cm^−1^.

Unlike suspended WTe_2_ sample, the heat dissipation of supported WTe_2_ flake should be described with a different model [40]
(8)$$\nabla \left(\kappa_{\|}\nabla T\right)+P-\kappa_{⊥}\left(T-T_{0}\right)=0,$$
where κ_‖_ is the in-plane thermal conductivity along the direction parallel to the current; κ_⊥_ is that along the direction perpendicular to the current; P is the power density per channel length; and T_0_ is the temperature when the bias voltage is 0 V. Since κ_‖_ is a constant, the equation above can be simplified to $T=T_{0}+P/\kappa_{⊥}(1–1/\cos h(L/L_{H}))$, where L_H_ is the thermal healing length and L is the channel length of the WTe_2_ device. Considering L»L_H_, it can be rewritten as the following equation [40]:(9)$$T=T_{0}+P/\kappa_{⊥}.$$

The lattice mismatch between WTe_2_ flake and SiO_2_/Si substrate may diminish interfacial heat dissipation [40,46], and Chen et al. suggest that the heat dissipation towards substrate may not enhance the thermal anisotropy ratio for supported samples [27]. Therefore, in-plane thermal dissipation is considered to be dominant here. Appendix A shows the schematic of in-plane thermal dissipation in the WTe_2_ device. Joule heat is generated when the current flows along the armchair direction of the WTe_2_ device and is then dissipated preferentially along the zigzag direction with greater thermal conductivity. For the in-plane heat dissipation of the WTe_2_ device, the effect of the contact resistance (R_c_) between WTe_2_ flake and electrodes on the electric power needs to be considered. When the channel length of a WTe_2_ device is 10 μm, the power is dissipated mostly along the channel (P_ch_ = P_t_ − I^2^R_c_ = 58%P_t_ [40], where P_ch_ means channel power, P_t_ means total power). Therefore, the power density in Equation (9) is mainly the channel power density P_ch_. According to temperature-dependent and bias voltage-dependent Raman spectra, we can calculate the temperature variation caused by channel power density for supported WTe_2_ flake.

Figure 4a,b shows the temperature as a function of channel power density for supported WTe_2_ flake. The extracted in-plane thermal conductivity for supported WTe_2_ flake is κ_zigzag_ = 62.17 ± 0.36 W·m^−1^·K^−1^ and κ_armchair_ = 32.93 ± 0.82 W·m^−1^·K^−1^ through Equation (9). Ma et al. found that the thermal conductivity of monolayer WTe_2_ sample along different directions was 20 and 9 W·m^−1^·K^−1^ [47], respectively, and it was previously reported that the thermal conductivity of 50 ± 2.8 nm-thick WTe_2_ sample supported on SiO_2_/Si substrate can reach 28 ± 11.41 W·m^−1^·K^−1^ [48], which are within the same order of magnitude as our results. Considering that the low-frequency phonons mainly contribute to the thermal conductivity [28] and that the thermal conductivity along the zigzag direction of WTe_2_ flake is larger. More importantly, the thermal anisotropy ratio of supported WTe_2_ flake (κ_zigzag/_κ_armchair_ ≈ 1.89) is about 1.7 times that of suspended WTe_2_ flake (κ_zigzag/_κ_armchair_ ≈ 1.09). It is inferred that the low-frequency phonons with anisotropy [47] may be affected when WTe_2_ flake is supported on silicon substrate, allowing the variation of the thermal anisotropy ratio. Furthermore, since the mechanical properties of materials will be affected when a mechanical force is applied [29], it is possible that the interaction between silicon substrate and WTe_2_ flake has affected the thermal transport properties in an uneven manner, and the examined thermal anisotropy has changed accordingly. Appendix A shows that the enhanced thermal anisotropy caused by substrate coupling can also be observed in BP sheets. The thermal anisotropy ratio of supported BP sheets has been improved and is almost unaffected by the heat shunting along the substrate [27], pointing out that substrate coupling can modulate the thermal anisotropy ratio of anisotropic 2D materials and inspire advanced device design.

## 4. Conclusions

In summary, the thermal transport behavior of 50 nm-thick suspended and supported WTe_2_ flakes along zigzag/armchair axes was investigated by Raman thermometry, demonstrating that substrate coupling could improve the thermal anisotropy of WTe_2_ flake to a great extent. The thermal conductivity (κ) of suspended WTe_2_ flake is κ_zigzag_ = 4.45 W·m^−1^·K^−1^ and κ_armchair_ = 4.10 W·m^−1^·K^−1^. For WTe_2_ device supported on SiO_2_/Si substrate, the thermal conductivity changes to κ_zigzag_ = 62.17 W·m^−1^·K^−1^ and κ_armchair_ = 32.93 W·m^−1^·K^−1^. When supported on a silicon substrate, the low-frequency phonons and mechanical properties that contribute to thermal dissipation may have anisotropic effects on the different lattice orientations of WTe_2_ flake, generating a higher thermal anisotropy ratio of supported WTe_2_ flake (about 1.7 times that of suspended WTe_2_ flake). Our work reveals the anisotropic thermal dissipation properties of multilayer WTe_2_ devices supported on silicon substrates, which may be useful for the application of WTe_2_ and other 2D anisotropic materials in nanodevices.

## Figures and Tables

**Figure 1 nanomaterials-13-01817-f001:**
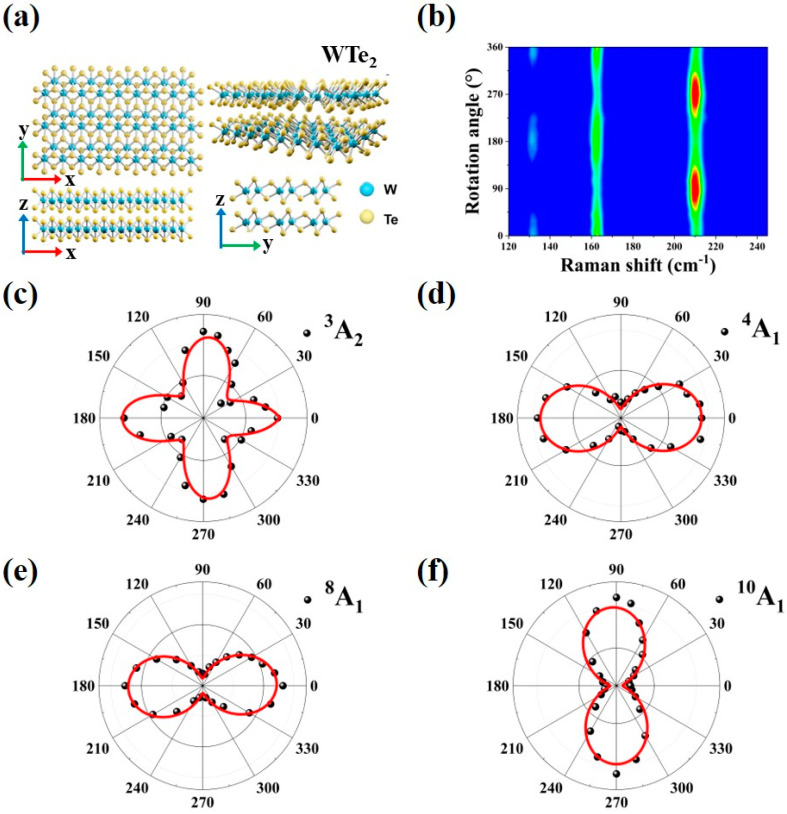
The crystal structure and angle-dependent Raman spectra of WTe_2_. (**a**) Top view and side view of the WTe_2_ crystal structure. (**b**) Contour map of angle-dependent Raman intensity for three A_1_ modes under parallel-polarized configuration. (**c**–**f**) Polar plots and fit curves of angle-dependent Raman intensity for ^3^A_2_, ^4^A_1_, ^8^A_1_, and ^10^A_1_ peaks.

**Figure 2 nanomaterials-13-01817-f002:**
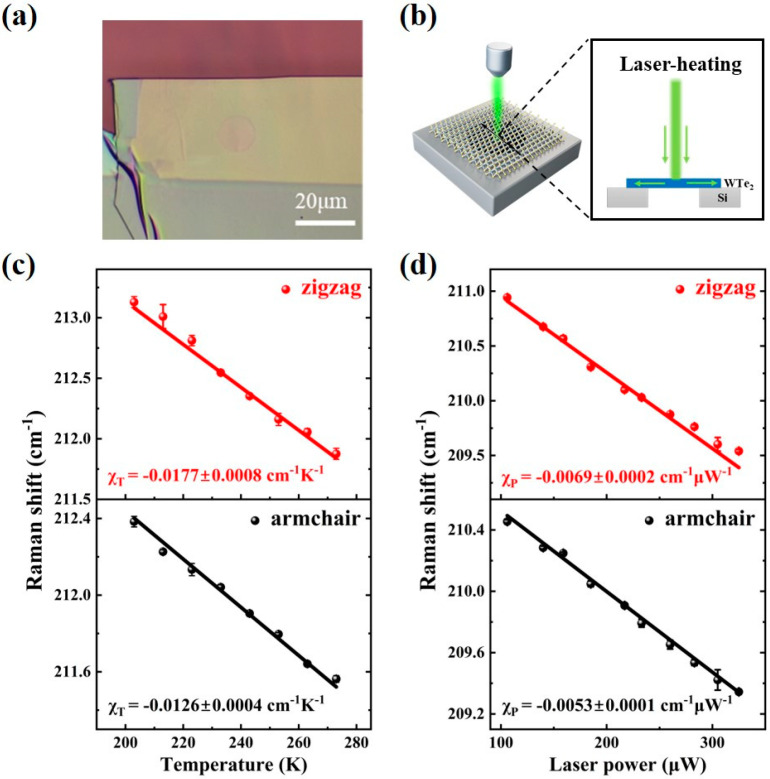
Temperature- and laser power-dependent in situ Raman spectra of suspended WTe_2_ sample. (**a**) Optical microscopy image of suspended WTe_2_ flake. (**b**) Schematic diagram of the experimental setup for Raman thermometry with laser-heating (the inset is the laser-heating process of suspended WTe_2_ flake). (**c**,**d**) The linear fitting of the ^10^A_1_ Raman peak position as a function of temperature (**c**) and laser power (**d**) along zigzag and armchair directions.

**Figure 3 nanomaterials-13-01817-f003:**
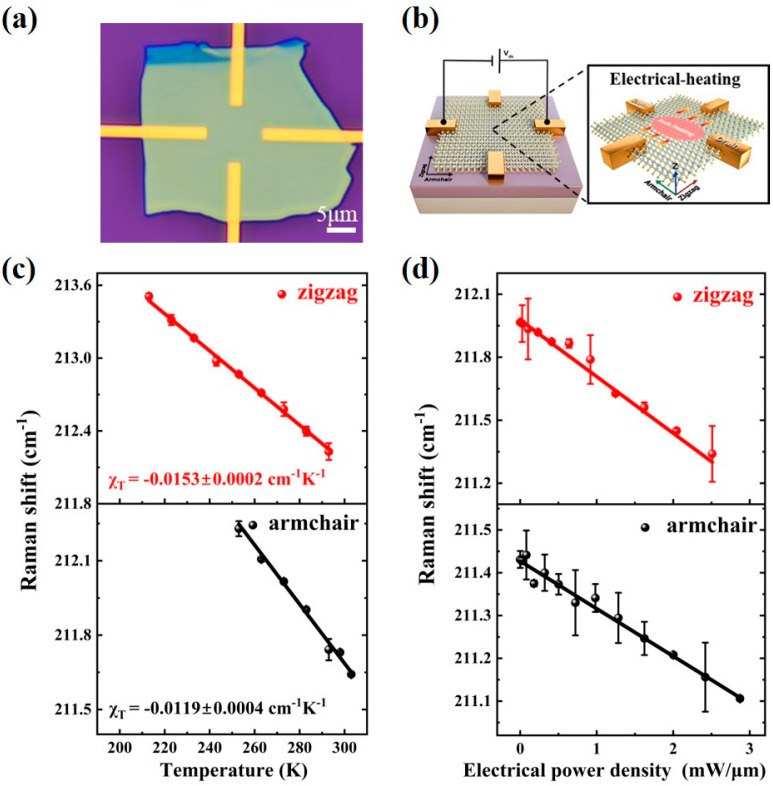
Temperature- and bias voltage-dependent in situ Raman spectra of supported WTe_2_ sample (WTe_2_ device supported on SiO_2_/Si substrate). (**a**) Optical image of supported WTe_2_ flake. (**b**) Schematic diagram of Raman thermometry with electrical heating (the inset depicts heat dissipation process in supported WTe_2_ flake). (**c**,**d**) The linear fitting of the ^10^A_1_ Raman peak position as a function of temperature (**c**) and electrical power density (**d**) along both zigzag and armchair directions.

**Figure 4 nanomaterials-13-01817-f004:**
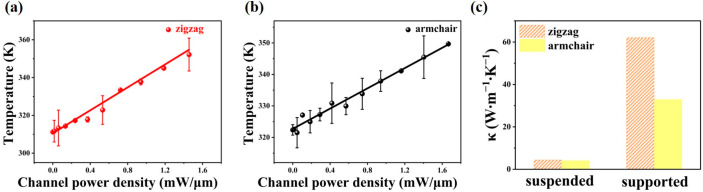
The effect of substrate coupling on thermal anisotropy. The fitting curve of temperature versus channel power density along zigzag (**a**) and armchair (**b**) directions for supported WTe_2_ device. (**c**) In-plane thermal conductivity of suspended and supported WTe_2_ samples along zigzag and armchair directions.

## Data Availability

The data presented in this study are available on request from the corresponding authors.

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
