# Peer review of "Improved Thermal Anisotropy of Multi-Layer Tungsten Telluride on Silicon Substrate"

_nanomaterials, 2023, doi:10.3390/nano13121817_

Round 1

Reviewer 1 Report (Previous Reviewer 1)

Dear Authors!

Thank you for your work on the manuscript under consideration. Still, I do not understand how exactly the anisotropy of thermal conductivity can be deduced from presented "laser heating" experiments. I suspect that the work has a major flaw in the analisys at this point, which is definitely critical.

Though minor language issues are present in the manuscript, the text is completely understandable.

Author Response

Reviewer 2 Report (Previous Reviewer 2)

The authors addressed carefully my remarks. The paper is publishable.

English is OK.

Author Response

Reviewer 3 Report (New Reviewer)

In this manuscript, the authors report an experimental measure of the anisotropy of thermal conductivity in a 50nm thick WTe2 layer, suspended and supported on top of SiO2/Si substrate.

The authors measure that the zig-zag is greater than the armchair thermal conductivity, and that the anisotropy is larger in the presence of the substrate than in the suspended case.

The paper is well written, the results and science are sound and the methodology is well explained.

Is only, I recommend the authors to explain the meaning of the terms shown in equations placed at lines 140 and 147 of the manuscript just after the equations even when the meaning of the different coefficients is given later in the text, and to add parenthesis to all the terms in the denominator of the fraction to remote any chance of ambiguity.

Therefore, I recommend this manuscript for publication.

Round 2

Reviewer 1 Report (Previous Reviewer 1)

Dear Authors!

Thank you once more for your work on the manuscript! Though certain question still have to be addressed, it might be practical to publish the paper "as is". After all, the best figure of merit for a research article is the responce of the journal's audience.

Sincerely yours, 

Reviewer

English is ok in the manuscript

This manuscript is a resubmission of an earlier submission. The following is a list of the peer review reports and author responses from that submission.

Round 1

Reviewer 1 Report

Dear Editor and Authors!

The manuscript under consideration is quite an interesting work to read; however, as a reviewer, I believe that it contains serious flaws and can not be published in its present state. Following the title, two key words in the article referring to thermal transport appear to be “anisotropy” and “improving”. Unfortunately, both are not presented correctly in the manuscript.

First, we refer to the results for suspended WTe2 sample (Fig.2 in the manuscript). Though Raman shifts given in Figs.2c and 2d are reasonable, is remains completely unclear how do the Authors evaluate the anisotropy of thermal transport: (i) the experimental setup itself is perfectly symmetrical and isotropic, and (ii) all anisotropic effects (if present) are already lost once the equation (6) is written down, since k does not depend on the direction. Thus, different k-values for “zigzag” and “armchair” may indeed be anticipated, but not extracted from existing experimental data.

Second, the results for supported WTe2 sample. Despite that the Authors provide refs. 50 and 54, it remains puzzling that vertical heat transport (through the substrate) is considered negligible and only in-plane heat dissipation is analyzed. In fact, the effect is neither mentioned nor explained in [50]. As a result, it is rather hard to believe in an order of magnitude improvement in lateral thermal conductivities due to somewhat modified phonon spectrum. Intuitively, noticeable phonon coupling should result in a noticeable vertical heat transport, which the Authors dismiss.

Minor comments may be addressed to experimental details, such as relevant temperature ranges (surprisingly, all studies are below room-temperature), base temperature / laser power in Figs.2c/2d, measurement technique and accuracy for the absorbance measurements (Fig.S4) and so on.

To conclude, in my opinion, the manuscript has to be completely reconsidered and rewritten, and my recommendation is to reject the work in its present state.

There are definitely certain grammar issues in the manuscript. However, it remains completely understandable and readable.

Reviewer 2 Report

The paper is devoted to the important topic. The results are novel. However the paper is written in a careless way and needs a serious revision.

Remarks

1. In the text: "The thermal conductivity  of suspended WTe2 flake is zigzag = 4.4 Wm−1K−1 and armchair = 4.1 Wm−1K−1. For WTe2 device supported on SiO2/Si substrate, the thermal conductivity changes to zigzag = 62 Wm−1K−1 and armchair = 33 Wm−1K−1".

What is the statistical scattering of the reported results? What was the statistical accuracy? Without the appropriate statistical treatment, the supplied results are senseless.

2. What is the statistical scattering of the results, represented in Figures 2-3? Without accurate and adequate statistical treatment no scientific paper should  be published.

3. It should be very useful to address also the thermal capacity and mechanical properties of the reported chalcogenide glasses, see:

Voronel A. et al., Mechanical and thermodynamic properties of infrared transparent low melting chalcogenide glass, Infrared Physics & Technology, 43 (6), 2002, Pages 397-399

The English is OK.